# Oxidized-LDL Deteriorated the Renal Residual Function and Parenchyma in CKD Rat through Upregulating Epithelial Mesenchymal Transition and Extracellular Matrix-Mediated Tubulointerstitial Fibrosis—Pharmacomodulation of Rosuvastatin

**DOI:** 10.3390/antiox11122465

**Published:** 2022-12-15

**Authors:** Pei-Hsun Sung, Ben-Chung Cheng, Tsuen-Wei Hsu, John Y Chiang, Hsin-Ju Chiang, Yi-Ling Chen, Chih-Chao Yang, Hon-Kan Yip

**Affiliations:** 1Division of Cardiology, Department of Internal Medicine, Kaohsiung Chang Gung Memorial Hospital, College of Medicine, Chang Gung University, Kaohsiung 83301, Taiwan; 2Institute for Translational Research in Biomedicine, Kaohsiung Chang Gung Memorial Hospital, Kaohsiung 83301, Taiwan; 3Center for Shockwave Medicine and Tissue Engineering, Kaohsiung Chang Gung Memorial Hospital, Kaohsiung 83301, Taiwan; 4Division of Nephrology, Department of Internal Medicine, Kaohsiung Chang Gung Memorial Hospital, College of Medicine, Chang Gung University, Kaohsiung 83301, Taiwan; 5Department of Computer Science and Engineering, National Sun Yat-sen University, Kaohsiung 804201, Taiwan; 6Department of Healthcare Administration and Medical Informatics, Kaohsiung Medical University, Kaohsiung 80708, Taiwan; 7Department of Obstetrics and Gynecology, Kaohsiung Chang Gung Memorial Hospital, College of Medicine, Chang Gung University, Kaohsiung 83301, Taiwan; 8School of Medicine, Chung Shan Medical University, Taichung 40201, Taiwan; 9Department of Medical Research, China Medical University Hospital, China Medical University, Taichung 40402, Taiwan; 10Department of Nursing, Asia University, Taichung 41354, Taiwan; 11Division of Cardiology, Department of Internal Medicine, Xiamen Chang Gung Hospital, Xiamen 361028, China

**Keywords:** chronic kidney disease, oxidized low-density lipoprotein, epithelial mesenchymal transition, oxidative stress

## Abstract

This study tested the hypothesis that intrarenal arterial transfusion of oxidized low-density lipoprotein (ox-LDL) jeopardized the residual renal function and kidney architecture in rat chronic kidney disease ((CKD), i.e., induced by 5/6 nephrectomy) that was reversed by rosuvastatin. Cell culture was categorized into A1 (NRK-52E cells), A2 (NRK-52E + TGF-β), A3 (NRK-52E + TGF-β + ox-LDL) and A4 (NRK-52E + TGF-β + ox-LD). The result of in vitro study showed that cell viability (at 24, 48 and 72 h), NRK-52E ox-LDL-uptake, protein expressions of epithelial–mesenchymal–transition (EMT) markers (i.e., p-Smad2/snail/α-SMA/FSP1) and cell migratory and wound healing capacities were significantly progressively increased from A1 to A4 (all *p* < 0.001). SD rats were categorized into group 1 (sham-operated control), group 2 (CKD), group 3 (CKD + ox-LDL/0.2 mg/rat at day 14 after CKD induction) and group 4 (CKD + ox-LDL-treated as group 3+ rosuvastatin/10 mg/kg/day by days 20 to 42 after CKD induction) and kidneys were harvested at day 42. The circulatory levels of BUN and creatinine, ratio of urine-protein to urine-creatinine and the protein expressions of the above-mentioned EMT, apoptotic (cleaved-caspase3/cleaved-PARP/mitochondrial-Bax) and oxidative-stress (NOX-1/NOX-2/oxidized-protein) markers were lowest in group 1, highest in group 3 and significantly higher in group 4 than in group 2 (all *p* < 0.0001). Histopathological findings demonstrated that the kidney injury score, fibrotic area and kidney injury molecule-1 (KIM-1) displayed an identical pattern, whereas the cellular expression of podocyte components (ZO-1/synaptopodin) exhibited an opposite pattern of EMT markers (all *p* < 0.0001). In conclusion, ox-LDL damaged the residual renal function and kidney ultrastructure in CKD mainly through augmenting oxidative stress, EMT and fibrosis that was remarkably reversed by rosuvastatin.

## 1. Introduction

There is regrettably lacking an effective therapy for preserving the residual renal function of chronic kidney disease (CKD), leading to not only a growing burden of healthcare costs [1] but also ultimately a result of end-stage renal disease (ESRD) [2] and unacceptably high morbidity and mortality [3]. Renal fibrosis has been identified as an inevitable process that frequently involves the initiation and evolution of CKD [2,3]. Currently, the mechanism of renal fibrosis is believed to be multifactorial and yet completely understood [4,5,6]. In fact, renal fibrosis is a complicated process that involves an interplay between epithelial cells, extracellular matrix, vascular cells, myofibroblasts, and immune cells [7]. As the process of renal fibrosis initiates, multiple physical changes occur between tubulointerstitial regions, including interstitial fibrosis and tubular atrophy [8]. The accumulated extracellular matrix (ECM) fibrils restrict the function of surrounding small vessels, resulting in reduced blood flow, decreased water motility, and delayed oxygen diffusion rate [9,10]. Therefore, ischemic tubular injury starts to trigger tubular cell apoptosis, leading to an increase of profibrotic stimuli to drive expandable renal fibrosis [9,10].

The epithelial–to-mesenchymal transition (EMT) is recognized as being essential for embryogenesis and tumor progression [11]. EMT is a cellular program by which epithelial cells lose their cell–cell adhesion and gain the properties of migration to possess the different fate in the mesenchymal type. It has been demonstrated that tubular epithelial cells undergo phenotypic conversion after treatment with transforming growth factor-beta (TGF-β) and the conversion is characterized by loss of epithelial proteins such as E-cadherin, and zonula occludens-1 (ZO-1), and gain of mesenchymal markers such as vimentin, alpha-smooth muscle actin (alpha-SMA), fibroblast-specific protein-1 (FSP1), and fibronectin [12,13]. Additionally, these alterations are usually accompanied by morphologic changes to a fibroblast-like appearance [14,15,16]. EMT-associated morphology and phenotype as well as EMT markers were detected in the tubular epithelium and peritubular interstitium after nephrectomy in rodent [17]. 

Oxidized low-density lipoprotein (LDL) is a potentially risky type of cholesterol for arterial atherosclerosis that is generated when normal LDL cholesterol is damaged by free radicals and can be taken up by cells through scavenger receptor-mediated endocytosis [18,19]. Interestingly, several studies have reported that oxidized LDL was involved in the EMT process through its receptor interactions to trigger a cascade of the signaling pathway [20,21,22,23] and specific miRNAs [24]. Additionally, it has been reported that the elevation of plasma oxidized LDL leads to early fibrosis in not only kidney parenchyma across the whole spectrum of CKD stages [25,26,27] but also cardiac myofibroblasts [28]. Furthermore, endothelial dysfunction and oxidative stress have been identified as commonly present in diabetic CKD and are strongly associated with unfavorable prognostic outcomes [29]. Moreover, CKD always elicits oxidative stress and inflammation, resulting in clinical adverse events [30].

Based on the above-mentioned issues [9,10,11,12,13,14,15,16,17,18,19,20,21,22,23,24,25,26,27,28,29,30], we tested the hypothesis that oxidized LDL might play an essential role in the deterioration of the residual renal function in setting of CKD through upregulation of renal tubular epithelial cells EMT process and renal fibrotic signaling pathway.

## 2. Materials and Methods

### 2.1. In Vitro Study Design

#### 2.1.1. To Elucidate the Impact of Oxidized LDL on Facilitating TGF-β Induced EMT Process in Cell Model

It is well recognized that the EMT process is mainly regulated by transforming growth factor beta (TGF-β). Thus, in our in vitro study, we utilized the TGF-β co-cultured with NRK-52E. Additionally, whether oxidized LDL would facilitate TGF-β to boost the progression of EMT was investigated.

#### 2.1.2. Cell Culture and Cell Grouping

NRK-52E cells (rat renal proximal tubular cell line) were purchased from the Bioresource Collection and Research Center (BCRC, Taipei, Taiwan) and cultured in DMEM(H) medium with 5% CFS. All cells were maintained at 37 °C under 5% CO_2_. For individual in the in vitro study, the cells were categorized into A1 (NRK-52E cell line), A2 (NRK-52E + TGF-β (5 ng/mL co-cultured for 48 h)), A3 (NRK-52E + TGF-β (5 ng/mL) + oxidized LDL (5 μg/mL co-cultured for 48 h)), and A4 (NRK-52E + TGF-β (5 ng/mL) + oxidized LDL (20 μg/mL co-cultured for 48 h)), respectively.

#### 2.1.3. Lipid Droplet Staining

To elucidate the uptake capacity of NRK-52E cells to ox-LDL, the Dil-labeled oxidized LDL (i.e., purchased from Invitrogen) was utilized to induce the accumulation of lipid droplets in NRK-52E cells. After TGF-β and oxidized LDL treatments, the lipid droplets within NRK-52E cells were identified and quantitated by using LipidTOX™ Green neutral lipid stain reagent (Invitrogen, Waltham, MA, USA, H34475).

#### 2.1.4. Assessment of Wound Healing Migratory Ability

To assess the impact of TGF-β and oxidized LDL treatments on enhancing the growth and migratory speed of NRK-52E cells, these cells were categorized into A1 to A4, respectively. To investigate the growth speed of NRK-52E cells, the cells were incubated in the disk and an artificial ditch was homogeneously created at the baseline (i.e., at 0 h) and the cells were incubated on both sides separately. By 24 h after cell culture, we measured the residual confluence area, defined as a wound healing process (%) that was estimated by (initial area − final area)/(initial area).

To determine the migratory ability, the NRK-52E cells were cultured in the Transwell. By 24 h after cell culture, the membrane was removed and the migrated cells in the bottom side of the Transwell were carefully counted for the analysis of migratory ability.

### 2.2. In Vivo Study Design

#### 2.2.1. Animal Model of 5/6 Nephrectomy for CKD Induction

Pathogen-free, adult male Sprague Dawley (SD) rats, weighting around 300–320 g (Charles River Technology, BioLASCO Taiwan Co., Ltd., Taipei, Taiwan), were utilized in the present study. The procedure and protocol of 5/6 nephrectomy for induction of the CKD animal model were based on our recent reports [31,32]. In detail, all the animals were anesthetized with inhalational 2.0% isoflurane, placed supine on a warming pad at 37 °C for midline laparotomies. Sham-operated rats received laparotomy only, while CKD was induced by right nephrectomy plus arterial ligation of upper two-third (upper and middle poles) blood supplies of the left kidney (i.e., by leaving lower one-third (lower pole) kidney with normal blood supply). Such a CKD model allowed preservation of a limited amount of function in renal parenchyma and offered a simulation of CKD setting. After 5/6 nephrectomy, oxidized LDL was given by intra-renal arterial administration. After 14 days of CKD induction, rosuvastatin (10 mg/kg/day) was given to the lower serum level of oxidized LDL. By 42 days after 5/6 nephrectomy surgery, peripheral blood was collected for the circulatory levels of blood urine nitrogen (BUN) and creatinine. The animals in each group were euthanized and the kidneys were harvested for individual examination.

The animals (n = 32) were equally categorized into group 1 (sham-operated control), group 2 (CKD + 0.5cc normal saline by intrarenal arterial administration by day 14 after CKD induction), group 3 (CKD + oxidized LDL/0.2 mg/rat by intrarenal artery administration once by day 14 after CKD induction), and group 4 (CKD + oxidized LDL/0.2 mg/rat by intrarenal artery administration once by day 14 after CKD induction + rosuvastatin/10 mg/kg/day since day 20 to day 42 after CKD induction) and the kidney in each animal was harvested at day 42. Because the company (i.e., Charles River Technology, BioLASCO Taiwan Co., Ltd., Taipei, Taiwan) could not provide the female animals for this study, we only utilized male animals in the present study.

#### 2.2.2. Examination of Renal Function Parameters

To elucidate if the CKD animal model was satisfactorily achieved, blood samples were serially drawn prior to and after the CKD induction (i.e., before and by days 14, 28, and 42 prior to the animals were sacrificed). Plasma levels of creatinine and BUN were analyzed by utilizing the standard laboratory method.

#### 2.2.3. Collection of 24 h Urine for the Ratio of Urine Protein to Urine Creatinine (RuPr/uCr) Prior to and by Day 42 after CKD Procedure

The procedure and protocol were based on our recent study [32]. To collect 24 h urine for the RuPr/uCr, each animal in every group was placed in a metabolic cage (DXL-D, Suzhou Fengshi Laboratory Animal Equipment Co., Ltd., Suzhou, China) for 24 h, and food and water were freely accessed for the animals. A total of 24 h urine was collected from each animal prior, and by day 42 after CKD, was conducted for assessing the RuPr/uCr.

#### 2.2.4. Histopathological Assessment of Fibrotic Area

Masson’s trichrome stain was applied for the determination of fibrotic feature in kidney specimens. Three 4 μm-thick serial fragments of kidney tissue were acquired by Cryostat (Leica CM3050S). The fibrosis region was calculated using ImageJ software. Three selected fragments were quantified for each animal. Three randomly chosen HPFs (100×) were analyzed in each fragment.

#### 2.2.5. Histopathologic Scoring of Kidney Damage by Day 42 after CKD Procedure

The pathological assessment of kidney injury scoring was determined in a blinded manner that has been addressed by our previous studies [32,33]. In detail, the harvested left kidney samples from each group of the animals were fixed in 10% buffered formalin, embedded in paraffin, at 4 µm and stained specimen (hematoxylin and eosin; H & E) for light microscopy. The scoring indicated the degree of tubular necrosis, loss of brush border, cast formation, tubular dilatation, and Bowmen’s capsule dilatation in 10 randomly selected, non-overlapping fields (200×) for each animal listing as: 0 (none), 1 (≤10%), 2 (11–25%), 3 (26–45%), 4 (46–75%), and 5 (≥76%).

#### 2.2.6. Western Blot Assessment of Left Kidney Specimens

The methodology was based on our recent studies [32,33]. Briefly, primary antibodies against p-Smad2 (1:1000, Cell Signaling, Danvers, MA, USA), p-Smad2 (1:1000, Cell Signaling), Snail (1:1000, Cell Signaling), α-SMA (1:5000, Sigma, MA, USA), ferroptosis suppressor protein 1 (FSP1) (1:1000, Cell Signaling), E-cadherin (1:1000, Abcam, Cambridge, UK), Laminin (1:1000, Novus Biologicals, Centennial, CO, USA), Elastin (1:1000, Affinity Biosciences, Cincinnati, OH, USA), Collagen type I (1:5000, Sigma), fibronectin (1:1000, Abcam), transforming growth factor (TGF)-β1 (1:3000, Abcam), cleaved-Caspase3 (1:1000, Cell Signaling), cleaved-PARP (1:1000, Cell Signaling), mitochondrial Bax (1:1000, Abcam), NOX-1 (1:1000, Sigma), NOX-2 (1:1000, Sigma), oxidized protein (1:100, Millipore, Burlington, MA, USA), Vimentin (1:1000, Cell Signaling), matrix metalloproteinase (MMP)2 (1:1000, Cell Signaling), MMP9 (1:1000, Abcam), Actin (1:10,000, Chemicon, Tokyo, Japan), and COXIV (1:10,000, Abcam) were used. Signals were detected with horseradish peroxidase (HRP)-conjugated goat anti-mouse, goat anti-rat, or goat anti-rabbit IgG.

Immunoreactive bands were visualized by enhanced chemiluminescence (ECL; Amersham Biosciences, Buckinghamshire, UK), which were then exposed to Biomax L film (Kodak). For quantification, ECL signals were digitized using Labwork software (UVP). For oxyblot protein analysis, a standard control was loaded on each gel.

#### 2.2.7. Immunofluorescent (IF) Study

The procedure and protocol for IF examination have been described by our recent investigations [32,33]. Briefly, IF staining was conducted for the assessment of synaptopodin (1:500, Santa Cruz, CA, USA), kidney injury molecule (KIM)-1 (1:400, Novus Biologicals, Centennial, CO, USA), and zonula occludens-1 (ZO-1) (1:200, Novus). The respective primary antibody was used with irrelevant antibodies as controls. Three sections of kidney specimens were analyzed in each rat. For quantification, three randomly selected HPFs (×200 for IF study) were analyzed in each section. The mean number per HPF for each animal was then determined by summation of all numbers divided by 9. An IF-based scoring system was adopted for semi-quantitative analysis of KIM-1 in the kidney as a percentage of positive cells in a blinded fashion (score of positively-stained cells: 0 = negative staining; 1 = <15%; 2 = 15–25%; 3 = 26–50%; 4 = 51–75%; 5 = 76–100% per high-power filed (HPF)). Additionally, the fluorescent intensity was utilized for the analysis of the ZO-1 and synaptopodin expression. 

#### 2.2.8. Statistical Analysis

Quantitative data are expressed as mean ± standard deviation. Statistical analyses were performed using SAS statistical software for Windows version 8.2 (SAS Institute, Cary, NC, USA). ANOVA was conducted, followed by Bonferroni multiple comparison post hoc test for comparing variables among groups. A probability value <0.05 was considered statistically significant.

## 3. Results

### 3.1. Preliminary Results for Pointing out a Corrective Direction of the Study (Appendix A)

First, before carrying on the present study, we tried to gather some preliminary data to support our hypothesis. Thus, we utilized the NRK-52E cells (i.e., rat proximal renal tubular cell line) which were categorized into G1 (i.e., NRK-52E), G2 (NRK-52E + TGF-β (5 ng/mL)), G3 (NRK-52E + TGF-β + oxidized LDL (5 μg/mL)) and G4 (NRK-52E + TGF-β + oxidized LDL (20 μg/mL)), respectively. The result showed that the protein expressions of phosphorylated (p)-Smad2, snail, alpha smooth muscle actin (α-SMA), and fibroblast-specific protein 1 (Fsp1), four biomarkers of EMT, were notably progressively increased from G1 to G4 (Appendix A). These findings provided essential information to schematically illustrate the simplified underlying mechanism of EMT process in which oxidized LDL facilitated TGF-β stimulation in renal tubular cells of CKD (Figure 4).

### 3.2. The Protein Expressions of EMT Biomarkers in NRK-52E Cell Line Undergoing the TGF-β and Oxidized LDL Treatments (Figure 1)

To test whether oxidized LDL treatment would facilitate the TGF-β to enhance the protein expressions of EMT biomarkers, the NRK-52E cells were categorized into A1 (NRK-52E), A2 (NRK-52E + TGF-β (5 ng/mL)), A3 (NRK-52E + TGF-β + oxidized LDL (5 μg/mL)) and A4 (NRK-52E + TGF-β + oxidized LDL (20 μg/mL)). The result demonstrated that the protein expressions of p-Smad2, snail, α-SMA, and Fsp1, four indicators of EMT, were significantly and progressively increased, whereas the protein expression of E-cadherin, an indicator of endothelial cell (EC) marker, was significantly and progressively reduced from A1 to A4. Additionally, the protein expressions of collagen type I, laminin, elastin, and fibronectin, four indices of extracellular proteins/fibrosis markers, exhibited an identical pattern of EMT biomarkers. These findings implicated that oxidized LDL treatment played a fundamental role in upregulation of EMT process with dose-dependent effect in renal tubular epithelial cells. Additionally, oxidized LDL, offered a synergic effect with TGF-β to augment the EMT process. Our findings could, at least in part, explain why fibrosis and perdition of renal function were commonly found in advanced CKD patients.

### 3.3. Cellular Levels of Fibrosis/ECM and Kidney Damaged Biomarkers (Figure 2)

It is well known that fibrosis/ECM is one of most common pathological findings in setting of CKD (referred to Figure 4) or after acute kidney injury. To verify whether oxidized LDL treatment would facilitate the TGF-β to upregulate the cellular expression of fibrosis and the renal tubular damage, the immunofluorescent (IF) microscopic finding was utilized in the present study with the conditions listed in Figure 1. As we expected, the IF microscope demonstrated that the cellular expressions of laminin, fibronectin, and collagen I, three extracellular matrix (ECM)/fibrosis markers, and cellular expression of KIM-1, an indicator of kidney damage, were significantly and progressively increased from A1 to A4. In this way, our results explicitly proved the aforementioned hypothesis, implying that histopathological features of CKD are rather complicated.

### 3.4. Morphological Features of NRK-52E Cells Followed by TGF-β and Oxidized LDL Stimulation (Appendix A)

To clarify morphological features of NRK-52E cells after TGF-β and oxidized LDL treatments, the NRK-52E cells were categorized into A1 to A4, i.e., as listed in Figure 1. The result showed that the phenotype of NRK-52E cells was identified to alter from epithelial-like phenotype to spindle-shape counterpart (i.e., indicated mesenchymal-like cells) after 7-day cell culture, especially more prominent in A4 group, implying these data once again proved that oxidized LDL would induce EMT process in renal tubular epithelial cells.

### 3.5. Lipid Droplets in NRK-52E Cells after Oxidized LDL Treatment and Inflammatory Cells Enhanced Extracellular Matrix (ECM) Production by NRK-52E Cells (Appendix A)

To clarify whether the uptake of oxidized LDL would be enhanced in NRK-52E cells, the IF microscope was utilized with the conditions listed in Figure 1. The result demonstrated that the number of lipid droplets within the cytoplasm of NRK-52E cells, i.e., an index of endocytosis of oxidized LDL, was significantly and progressively increased from A1 to A4, suggesting that NRK-52E cells had intrinsic capacity of uptake of oxidized LDL. 

It is well recognized that inflammatory reaction was commonly upregulated in kidney parenchyma in setting of CKD. Thus, we wanted to verify that inflammatory cells infiltration in renal tubules would enhance the production of ECM, i.e., an indirect biomarker of fibrotic upregulation, by renal tubular cells, and therefore, the NRK-52E cells were categorized into B1 (NRK-52E only) and B2 (NRK-52E + LPS-treated RAW 264.7 cells). The results of cellular levels and protein levels of laminin, fibronectin, and collagen I, three indicators of ECM, were significantly increased in B2 by comparison with B1. The findings were consistent with our hypothesis.

### 3.6. Impact of Synergic Effect of TGF-β and Oxidized LDL on Wound Healing Process, Migratory Assay, and Cell Viability (Figure 3)

Undoubtedly, oxidative stress/free radical and TGF-β were two cardinal factors participating in the fundamental mechanism of CKD in human being. To verify the impact of synergic effect of TGF-β and oxidized LDL on accelerating the wound healing process and migratory assay due to the EMT effect (i.e., loss of cell–cell contact, resulting in acceleration of migratory ability), the NRK-52E cells were categorized into A1 to A4, as indicated in Figure 1. The result showed that the cell migratory ability and wound healing process were significantly and progressively increased from A1 to A4, suggesting TGF-β and oxidized LDL enhanced EMT process, as a consequence of speed-up of the cell migratory ability and wound healing process.

Identically, the MTT assay demonstrated that the cell viability, indicator of ability of cell proliferation, expressed a similar pattern of wound healing process among the groups. The above finding indicated that TGF-β was significantly upregulated, and combined TGF-β and oxidized LDL, especially in higher dose of oxidized LDL, further significantly upregulated the cell proliferation. Our findings, once again, proved that oxidized-LDL and TGF-β were two essential biological factors participating in cell biological change of tubular epithelia cells to mesenchymal function and phenotype, i.e., a distinctive phenomenon of EMT process of the renal epithelial cells.

### 3.7. The Mechanism of Oxidized LDL Boosting TGF-β on the EMT Process in Renal Tubular Cells (Figure 4)

Based on the results of the present in vitro studies, we schematically proposed the underling mechanism of oxidized LDL boosting TGF-β on the EMT process in renal tubular cells. We highly speculated that the oxidized LDL cooperated with inflammatory cells, such as macrophages, to accelerate the process of EMT from renal tubular epithelial cells. In this way, the integrity of cell–cell contact would be weakened through undergoing EMT. After loss of cell–cell contact, pro-fibrotic stimuli would induce the deposition of ECM. Then, the accumulation of ECM fibrils acted as an obstacle to compress nearby capillaries, resulting in reduced blood flow and decreased oxygen delivery, causing a progressive kidney ischemic injury that would lead to initiation and propagation of renal fibrosis. This proposed mechanism has drawn our interest in the relationship between the level of oxidized LDL and the renal fibrosis. To further verify this issue, we designed an animal study.

### 3.8. The Time Courses of Circulating Levels of BUN and Creatinine and the Ratio of Urine Protein to Urine Creatinine (Figure 5)

At baseline prior to CKD induction, the circulatory levels of BUN and creatinine, and the ratio of urine protein to urine creatinine, did not differ among groups 1 (sham-operated control), 2 (CKD + 0.5cc normal saline), 3 (CKD + oxidized LDL), and 4 (CKD + oxidized LDL + rosuvastatin). Rosuvastatin acted as a lipid-lowering agent for the treatment of LDL elevation. By day 14 after CKD induction, these parameters were significantly lower in group 1 than in groups 2 to 4, but they showed no difference among groups 2 to 4. However, by days 28 and 42 after CKD induction, these parameters were significantly higher in group 3 than in other groups, significantly higher in groups 2 and 4 than in group 1, but these parameters did not differ between groups 2 and 4. These findings implicated that oxidized LDL notably deteriorated the residual renal function that could be remarkably reversed by rosuvastatin treatment, highlighting that the statin treatment sh ould be strongly recommended for those of CKD stage III and IV patients.

### 3.9. Protein Expressions of EMT Biomarkers in Kidney Parenchyma by Day 42 after CKD Induction (Figure 6)

To elucidate whether EMT biomarkers would be suppressed by rosuvastatin treatment, the Western blot analysis was applied. The result showed that the protein expressions of p-Smad2, snail, α-SMA, Fsp1, TGF-β and vimentin, six indices of EMT biomarkers, were significantly higher in group 3 than in groups 1, 2, and 4 and significantly higher in group 2 and 4 than in group 1, but they were similar between groups 2 and 4. On the other hand, the protein expression of E-cadherin, an indicator of the epithelial cell marker, displayed an opposite pattern of EMT markers among the groups. Our findings in the in vivo studies were not only consistent with the finding of the in vitro study but also ascertained the principal role of EMT on initiation and propagation of CKD.

### 3.10. Protein Expressions of Apoptotic and Oxidative Stress Biomarkers in Kidney Parenchyma by Day 42 after CKD Induction (Figure 7)

Further, we intended to delineate the impact of rosuvastatin therapy on alleviating apoptosis and oxidative stress. The results of Western blot analysis demonstrated that the protein expressions of mitochondrial Bax, cleaved caspase 3 and cleaved PARP, three indicators of apoptosis, were significantly higher in group 3 than in other three groups, significantly higher in groups 2 and 4 than in group 1, but they displayed a similar pattern between groups 2 and 4. Additionally, the protein expressions of NOX-1, NOX-2, and oxidized protein, three indices of oxidative stress, exhibited a similar pattern of apoptosis among the groups. The above findings suggested that there was a strong association between apoptosis and oxidative stress, and renal parenchymal disease that induced by oxidized LDL could effectively be suppressed by rosuvastatin.

### 3.11. Protein Expression of ECM in Kidney Parenchyma by Day 42 after CKD Induction (Figure 8)

Moreover, to determine the production of ECM in the kidney parenchyma, Western blot analysis was utilized once again. The result demonstrated that the protein expressions of MMP-2 and MMP-9, two indicator of proteolytic enzymes for accumulation of ECM in the extracellular space of kidney, were significantly higher in group 3 than in other groups, significantly higher in groups 2 and 4 than in group 1, but these parameters did not differ between groups 2 and 4. Additionally, the protein expressions of laminin, fibronectin, and collagen I, three indicators of ECM, displayed an identical pattern of MMPs among the groups, implying that inappropriate accumulation of ECM in kidney parenchyma of CKD could be a very common abnormal issue that compressed the space of renal tubules, i.e., the sparrow’s nest is occupied by a pigeon (referring to Figure 6). Fortunately, the production of ECM in kidney parenchyma was significantly inhibited by rosuvastatin treatment.

### 3.12. The Histopathological Analyses of Kidney Injury Score, Fibrosis, Kidney Injury Molecule, and Podocyte Components in Kidney Parenchyma by Day 42 after CKD Induction (Figure 9 and Figure 10)

Certainly, the ultrastructural integrity of glomeruli and podocyte components are strongly predictive of renal functional integrity and absence of proteinuria. Thus, we finally utilized the microscope to verify the severity of kidney parenchymal damage in the present study. The result showed that the kidney injury score (Figure 9) (i.e., summation of the parameters, including the grading of tubular necrosis, loss of brush border, cast formation, and Bowman’s capsule and tubular dilatation), fibrotic area (Figure 10) and KIM-1+ cells (Figure 10), were significantly higher in group 3 than in groups 1, 2, and 4, and significantly higher in groups 2 and 4 than in group 1, but they did not differ between groups 2 and 4. On the other hand, the cellular expressions of ZO-1 and synaptopodin (Figure 10), two podocyte components predominantly localized in glomeruli, exhibited an identical pattern of KIM-1 cells among the groups (Figure 10). Our finding, at least in part, provided useful information for explanation of mechanistic basis of proteinuria in CKD setting.

## 4. Discussion

This study, which investigated the impact of oxidized LDL and rosuvastatin on residual renal function and integrity of kidney parenchyma in setting of rodent CKD, has abundant striking implications. First, we successfully created a CKD animal model by which the renal function was markedly deteriorated by intrarenal arterial administration of oxidized LDL that was reproducibly utilized for the study. Second, the result of the present study clearly delineated that oxidized LDL augmented the expression of EMT markers and ECM deposition in the kidney parenchyma of rat CKD. Third, as compared with CKD + oxidized LDL animals, the residual renal function and renal architecture were notably preserved in CKD + oxidized LDL animals after receiving rosuvastatin. Finally, the results of the present study evidently appraised the underlying mechanism of oxidized LDL boosting TGF-β on the EMT process in the setting of rat CKD (refer to Figure 4).

Despite the inflammation, reactive oxygen species (ORS)/oxidative stress, fibrosis and immunogenicity having been extensively investigated as the fundamental etiologies for the initiation and propagation of the CKD, the impact of dyslipidemia, especially the fundamental role of oxidized LDL on outcomes of CKD, has not been clearly addressed to this day. Interestingly, an association between hypercholesterolemia and kidney injury has been recently established in an animal model study [34]. Additionally, some clinical studies have previously demonstrated that oxidized LDL was associated with the deterioration of kidney function and proteinuria beyond its properties of atherogenesis and vascular occlusion [35,36]. These clinical observational findings [34,35,36] implied that hypercholesterolemia, especially oxidized LDL, may play a crucial role in directly participating in the deterioration of renal function in human being. The most important finding in the present study was that the kidney injury score was significantly increased in CKD treated by oxidized LDL group than in that of CKD only. Additionally, not only the kidney injury score, but this study also further identified that the deterioration of residual renal function (i.e., increases in circulatory levels of BUN and creatinine and RuPr/uCr) was remarkably increased in CKD animals treated by oxidized LDL than in that of CKD only, suggesting oxidized LDL per se can cause both functional and structural renal damages. Furthermore, our findings, consistent with those of previous studies [34,35,36], obviously identified that oxidized LDL may play a paramount role on the deterioration of residual renal function in the setting of CKD. Of particular importance was that rosuvastatin therapy effectively protected the kidney against oxidized LDL damage in the CKD setting. It is well known that uremic toxic substances always elicit inflammation and ROS/oxidative stress [35,36], resulting in increasing both tissue and circulatory levels of oxidized LDL in CKD regardless of background total cholesterol level. Our results, therefore, pinpoint that for the patients with CKD, administering statin therapy could be a wise course of action for the prevention or treatment of further renal injury no matter how their circulatory oxidized LDL and total cholesterol levels are.

Surprisingly, when reviewing the literature [32,33,37], we always find that estimated glomerular filtration rate (eGFR) reduction and glomerular diseases (i.e., the glomerulus cell and apparatus damage), rather than the renal tubular epithelial cell (TEC) diseases, were considered as the principal factors of deterioration of CKD, implicating that the great important role of the renal TEC diseases on the progressive CKD had been regrettably neglected. Until recently, the accumulating number of investigations has shed light on the cardinal role of renal TECs in renal fibrosis and deterioration of renal function in the setting of CKD [37,38,39]. An essential finding in the present study was that the KIM-1 marker (i.e., a renal tubular damage marker) was notably increased in CKD animals and further, notably increased in oxidized LDL-treated CKD animals. Besides, the fibrosis area was identified predominantly localized in tubulointerstitial space, suggesting a close association between oxidized LDL and renal tubulointerstitial fibrosis. Moreover, when carefully examined the kidney injury score, we found that renal tubular damages, including tubular necrosis, loss of brush border, cast formation, and tubular dilatation, were much more frequently presented in oxidized LDL-treated CKD animals. Therefore, our findings not only were consistent with those of previous studies [37,38,39] but also indicated the relationship of oxidized LDL with chronic renal tubulointerstitial damage.

Intriguingly, previous studies have further emphasized that tubulointerstitial fibrosis is characterized by tubular atrophy and the accumulation of extracellular matrix (ECM) [7,38]. The cardinal finding in the present study was that MMPs activity and abundant accumulation of ECM along with generations of apoptosis and oxidative stress were found to remarkably increase in the CKD group and further remarkably increased in the oxidized treated CKD group as compared with those of the SC group. In this way, our findings echoed the findings of the previous studies [7,40].

EMT, characterized by acquiring primary mesenchymal markers of vimentin, alpha-SMA, FSP-1, collagen I, and fibronectin [16], is defined as the process that permits a polarized epithelial cell to adopt a mesenchymal-cell phenotype, resulting in increasing migratory ability, invasive behavior, and production of ECM components [41]. Of most importance was that these aforementioned EMT biomarkers [16] were observed to be significantly upregulated in NRK-52E cells (i.e., in vitro study) and in harvested kidney tissues (i.e., in vivo study) after oxidized LDL treatment, explaining that this switch (i.e., EMT process of renal epithelial cells) is also a potential source of fibroblasts and ECM deposition in renal tubulointerstitial space [7]. Additionally, the numbers of laminin, fibronectin, collagen, and lipid droplets cells were also identified to be substantially increased in oxidized LDL-treated NRK-52E cells (i.e., in vitro study). Our findings, in addition to supporting the results of previous studies [7,16,40,41], further let us delineate the mechanism of oxidized LDL boosting TGF-β on the EMT process in the setting of progressive CKD (Figure 4). Of distinctive importance was that all of the above-mentioned cellular-molecular perturbations were substantially suppressed by rosuvastatin treatment, i.e., not only in the in vitro but also in the in vivo experimental studies, implicating a mechanistical insight for renal protective effect of rosuvastatin against oxidized LDL in the setting of progressive CKD.

Our previous study has clearly demonstrated that the integrity of podocyte components was essential for avoiding the proteinuria [32,42]. In the present study, we found that the cellular expressions of podocyte components of ZO-1 and synaptopodin were notably reduced in CKD animals and further, remarkably reduced in oxidized LDL-treated CKD animals. Hence, our finding, besides supporting the findings of previous studies [32,42], could at least in part explain why the proteinuria was notably increased in CKD and further increased in oxidized LDL-treated CKD animals. Furthermore, these perturbations of podocytes and proteinuria were significantly reversed by rosuvastatin treatment in CKD + oxidized LDL animals, emphasizing again the importance of statin therapy for alleviation of progressive renal damage in the CKD setting.

Interestingly, a recent head-to-head comparative clinical trial has extensively investigated the impact of oxidative stress biomarkers on all-cause mortality in hemodialysis (HD) patients [43]. Disappointingly, the result showed that the oxidized LDL was not independently associated with all-cause mortality in the HD patients as compared to those with CKD in other stages [43]. When carefully delving deeper at the scenario of our animal study, one would easily realize that the animals were only in mild to moderate CKD stages rather than in the uremic stage, suggesting that the results of our study might be not extrapolatory into the clinical setting of end-stage renal disease.

Finally, in Figure 4, we summarized the fundamental steps in the underlying mechanism regarding how oxidized LDL induced deterioration of renal function in CKD. 

This study has limitations. First, without stepwise increasing of the dosage of oxidized LDL on the renal damage of CKD, we did not know whether the current dosage of oxidized LDL utilized in the present study was optimal or not. Second, we also did not know how high the circulatory level of oxidized LDL would damage the residual renal function and kidney architecture of CKD. Third, a group of CKD treated by rosuvastatin only (i.e., CKD + rosuvastatin) might be missed in the study, because the therapeutic effect of rosuvastatin has been keenly investigated by our previous studies [44,45].

## 5. Conclusions

The results of the present study demonstrated that oxidized LDL participated in deteriorating the residual renal function and kidney parenchyma in CKD through the EMT process from damaged renal tubular ECs, resulting in accumulating the ECM and augmenting the production of tubulointerstitial fibrosis. The detrimental effect of oxidized LDL on the progression of CKD could be reversed through statin therapy.

## Figures and Tables

**Figure 1 antioxidants-11-02465-f001:**
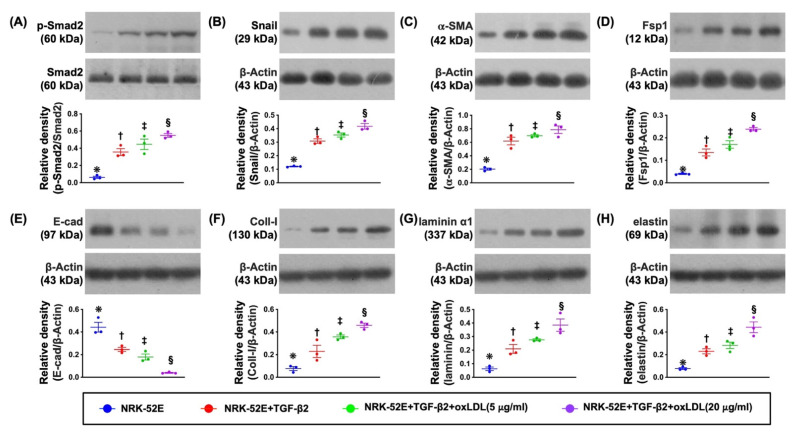
Protein expressions of EMT biomarkers in NRK-52E cells treated by TGF-β and oxidized LDL. (**A**) Protein expression of phosphorylated (p)-Smad2, * vs. other groups with different symbols (†, ‡, §), *p* < 0.001. (**B**) Protein expression of Snail, * vs. other groups with different symbols (†, ‡, §), *p* < 0.001. (**C**) Protein expression of alpha smooth actin (α-SMA), * vs. other groups with different symbols (†, ‡, §), *p* < 0.001. (**D**) Protein expression of fibroblast-specific protein 1 (Fsp1), * vs. other groups with different symbols (†, ‡, §), *p* < 0.001. (**E**) Protein expression of E-cadherin (E-cad), * vs. other groups with different symbols (†, ‡, §), *p* < 0.001. (**F**) Protein expression of collagen type I (Coll-I), * vs. other groups with different symbols (†, ‡, §), *p* < 0.001. (**G**) Protein expression of laminin, * vs. other groups with different symbols (†, ‡, §), *p* < 0.001. (**H**) Protein expression of elastin, * vs. other groups with different symbols (†, ‡, §), *p* < 0.001. All statistical analyses were performed by one-way ANOVA, followed by Bonferroni multiple comparison post hoc test (n = 3 for each group). Symbols (*, †, ‡, §) indicate significance for each other (at 0.05 level). TGF-β = transforming growth factor-beta; EMT = epithelial mesenchymal transition; LDL = low-density lipoprotein.

**Figure 2 antioxidants-11-02465-f002:**
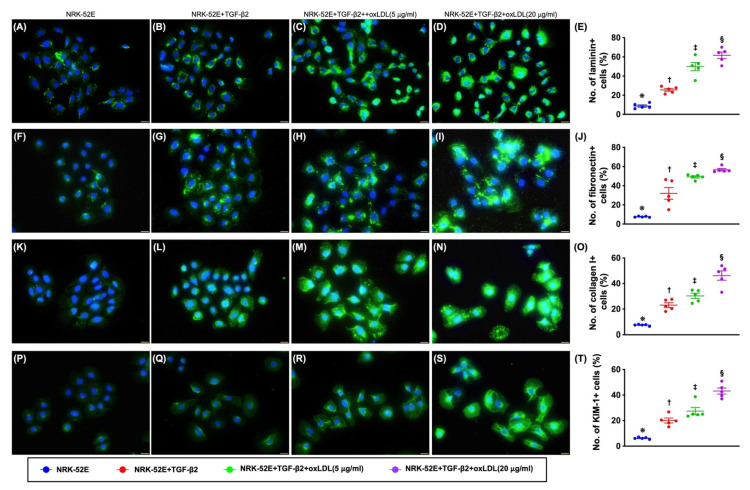
Cellular levels of fibrosis/ECM and kidney injury biomarkers in NRK-52E cells treated by TGF-β and oxidized LDL. (**A**–**D**) Illustrating the immunofluorescent (IF) stain of microscopic finding (400×) for identification of cellular expression of laminin (green color). (**E**) Analytical result of number (%) of laminin+ cells, * vs. other groups with different symbols (†, ‡, §), *p* < 0.001. (**F**–**I**) Illustrating the IF stain of microscopic finding (400×) for identification of cellular expression of fibronectin (green color). (**J**) Analytical result of number (%) of fibronectin+ cells, * vs. other groups with different symbols (†, ‡, §), *p* < 0.001. (**K**–**N**) Illustrating the IF stain of microscopic finding (400×) for identification of the expression of collagen I (green color). (**O**) Analytical result of number (%) of collagen I+ cells, * vs. other groups with different symbols (†, ‡, §), *p* < 0.001. (**P**–**S**) Illustrating the IF microscopic finding (400×) for identification of cellular expression of kidney injury molecule-1 (KIM-1) (green color). (**T**) Analytical result of number (%) of KIM-1+ cells, * vs. other groups with different symbols (†, ‡, §), *p* < 0.001. Scale bars in right lower corner represent 20 µm. All statistical analyses were performed by one-way ANOVA, followed by Bonferroni multiple comparison post hoc test (n = 5 for each group). Symbols (*, †, ‡, §) indicate significance for each other (at 0.05 level).

**Figure 3 antioxidants-11-02465-f003:**
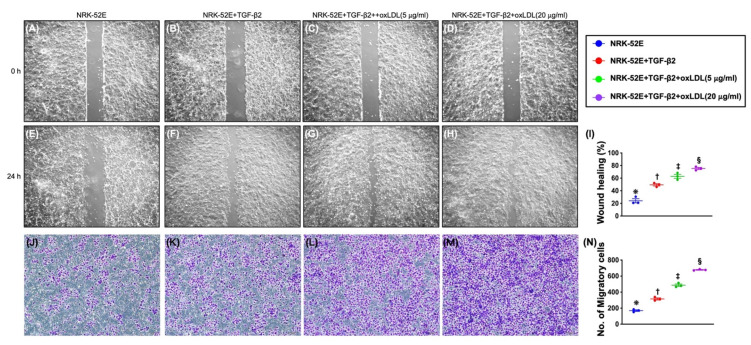
Impact of synergic effect of TGF-β and oxidized LDL on wound healing process and migratory assay of NRK-52E. (**A**–**D**) Illustrating the baseline (i.e., at 0 h) wound healing process. No difference in term of speed of the wound healing process. (**E**–**H**) Illustrating the 24 h morphological feature of wound healing process. (**I**) Analytical result of wound healing speed, * vs. other groups with different symbols (†, ‡, §), *p* < 0.0001. (**J**–**M**) Illustrating the microscopic finding (100×) for identification of cell migratory ability (pink-gray color) after 24 h incubation. (**N**) Analytical result of number of migratory cells, * vs. other groups with different symbols (†, ‡, §), *p* < 0.0001. Scale bars in right lower corner represent 100 µm. All statistical analyses were performed by one-way ANOVA, followed by Bonferroni multiple comparison post hoc test (n = 5 for each group). Symbols (*, †, ‡, §) indicate significance for each other (at 0.05 level).

**Figure 4 antioxidants-11-02465-f004:**
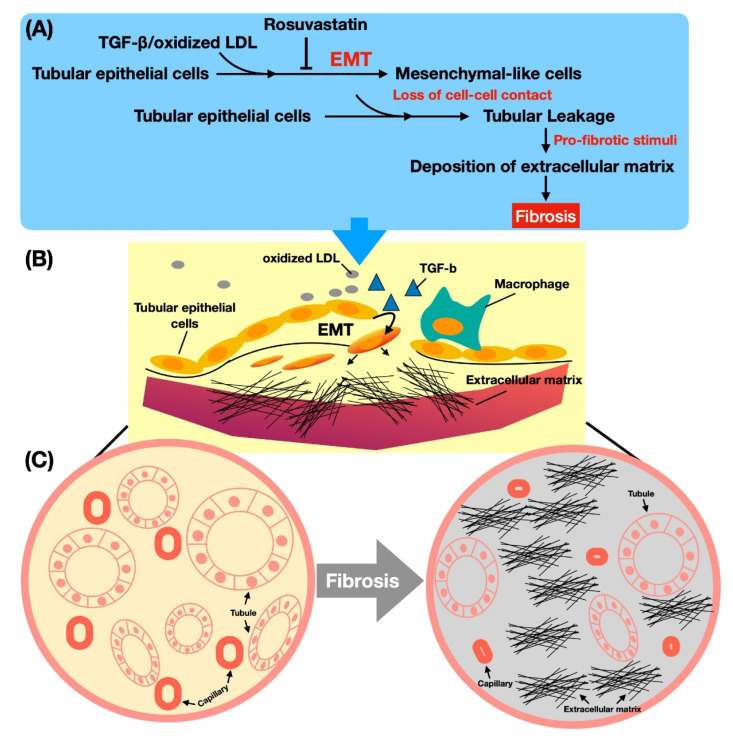
The mechanism of oxidized LDL boosting TGF-β on the EMT process in renal tubular cells. Based on the results of the in vitro studies, we schematically illustrate the underling mechanism of oxidized LDL boosting TGF-β on the EMT process in renal tubular cells, i.e., NRK-52E cells. Note that the **upper panel** (**A**) used the textual description to fully explain the pathological process of the renal tubular epithelial cells in the **middle panel** (**B**) after the oxidized LDL or TGF-β treatment. On the other hand, the **lower panel** (**C**) fundamentally concluded the final pathological outcomes of renal tubular epithelial cells after the oxidized LDL or TGF-β treatment. To illustrate this underlying mechanism of oxidized LDL or TGF-β induced renal tubular epithelial cells into EMT would lead to the reader easily understanding the impact of oxidized LD/TGF-β on the pathogenesis of EMT in the setting of CKD. A = upper panel; B = middle panel; C = lower panel.

**Figure 5 antioxidants-11-02465-f005:**
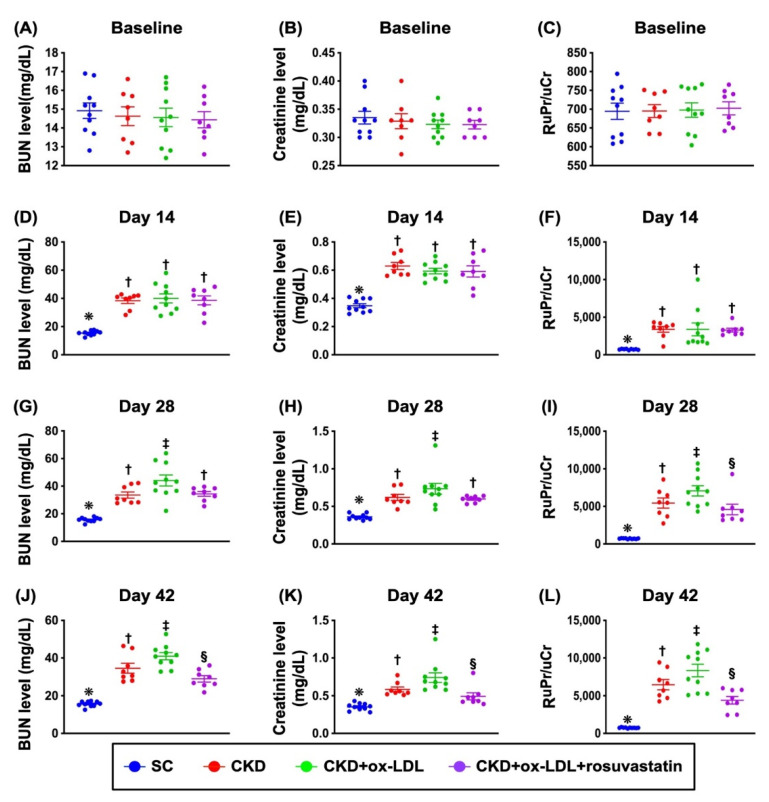
The time courses of circulating levels of BUN and creatinine and the ratio of urine protein to urine creatinine. (**A**) Baseline circulatory level of blood urine nitrogen (BUN), *p* > 0.5. (**B**) Baseline circulatory level of creatinine, *p* > 0.5. (**C**) Baseline ratio of urine protein to urine creatinine (RuPr/uCr), *p* > 0.5. (**D**) By day 14 after CKD induction, the circulatory BUN level, * vs. †, *p* < 0.0001. (**E**) By day 14 after CKD induction, the circulatory creatinine level, * vs. †, *p* < 0.0001. (**F**) By day 14 after CKD induction, the RuPr/uCr, * vs. †, *p* < 0.0001. (**G**) By day 28 after CKD induction, the circulatory BUN level, * vs. other groups with different symbols (†, ‡), *p* < 0.0001. (**H**) By day 28 after CKD induction, the circulatory creatinine level, * vs. other groups with different symbols (†, ‡), *p* < 0.0001. (**I**) By day 28 after CKD induction, the RuPr/uCr, * vs. other groups with different symbols (†, ‡, §), *p* < 0.0001. (**J**) By day 42 after CKD induction, the circulatory BUN level, * vs. other groups with different symbols (†, ‡, §), *p* < 0.0001. (**K**) By day 42 after CKD induction, the circulatory creatinine level, * vs. other groups with different symbols (†, ‡, §), *p* < 0.0001. (**L**) By day 42 after CKD induction, the RuPr/uCr, * vs. other groups with different symbols (†, ‡, §), *p* < 0.0001. All statistical analyses were performed by one-way ANOVA, followed by Bonferroni multiple comparison post hoc test (n = 8 for each group). Symbols (*, †, ‡, §) indicate significance for each other (at 0.05 level). SC = sham-operated control; CKD = chronic kidney disease; R = rosuvastatin.

**Figure 6 antioxidants-11-02465-f006:**
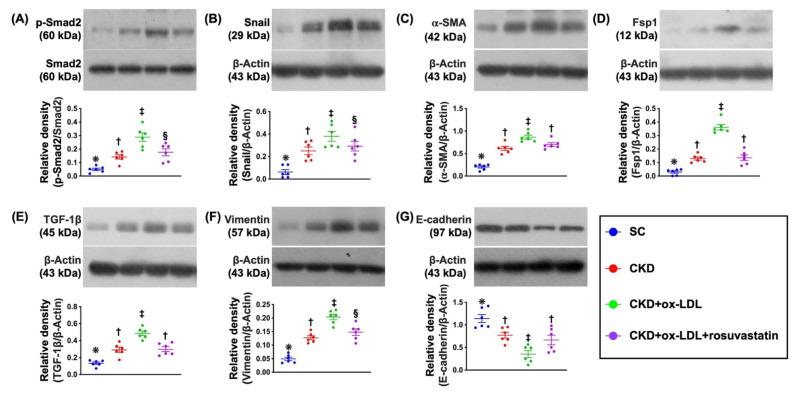
Protein expressions of EMT biomarkers in kidney parenchyma by day 42 after CKD induction. (**A**) Protein expression of phosphorylated (p)-Smad2, * vs. other groups with different symbols (†, ‡, §), *p* < 0.0001. (**B**) Protein expression of Snail, * vs. other groups with different symbols (†, ‡, §), *p* < 0.0001. (**C**) Protein expression of α-SMA, * vs. other groups with different symbols (†, ‡), *p* < 0.0001. (**D**) Protein expression of fibroblast-specific protein 1 (Fsp1), * vs. other groups with different symbols (†, ‡), *p* < 0.0001. (**E**) Protein expression of transforming growth factor (TGF)-1β, * vs. other groups with different symbols (†, ‡), *p* < 0.0001. (**F**) Protein expression of vimentin, * vs. other groups with different symbols (†, ‡, §), *p* < 0.0001. (**G**) Protein expression of E-cadherin, * vs. other groups with different symbols (†, ‡), *p* < 0.0001. All statistical analyses were performed by one-way ANOVA, followed by Bonferroni multiple comparison post hoc test (n = 6 for each group). Symbols (*, †, ‡, §) indicate significance for each other (at 0.05 level).

**Figure 7 antioxidants-11-02465-f007:**
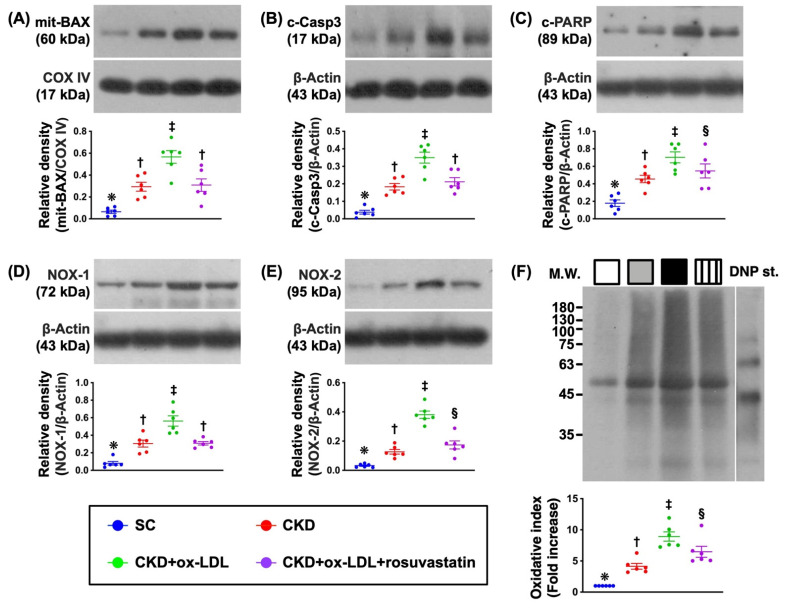
Protein expressions of apoptotic and oxidative stress biomarkers in kidney parenchyma by day 42 after CKD induction. (**A**) Protein expression of mitochondrial Bax (mit-Bax), * vs. other groups with different symbols (†, ‡), *p* < 0.0001. (**B**) Protein expression of cleaved caspase 3 (c-Casp3), * vs. other groups with different symbols (†, ‡), *p* < 0.0001. (**C**) Protein expression of cleaved poly (ADP-ribose) polymerase (c-PARP), * vs. other groups with different symbols (†, ‡, §), *p* < 0.0001. (**D**) Protein expression of NOX-1, * vs. other groups with different symbols (†, ‡), *p* < 0.0001. (**E**) Protein expression of NOX-2, * vs. other groups with different symbols (†, ‡, §), *p* < 0.0001. (**F**) The oxidized protein expression, * vs. other groups with different symbols (†, ‡, §), *p* < 0.0001 (Note: the left and right lanes shown on the upper panel represent protein molecular weight marker and control oxidized molecular protein standard, respectively). M.W. = molecular weight; DNP = 1–3 dinitrophenylhydrazone. All statistical analyses were performed by one-way ANOVA, followed by Bonferroni multiple comparison post hoc test (n = 6 for each group). Symbols (*, †, ‡, §) indicate significance for each other (at 0.05 level).

**Figure 8 antioxidants-11-02465-f008:**
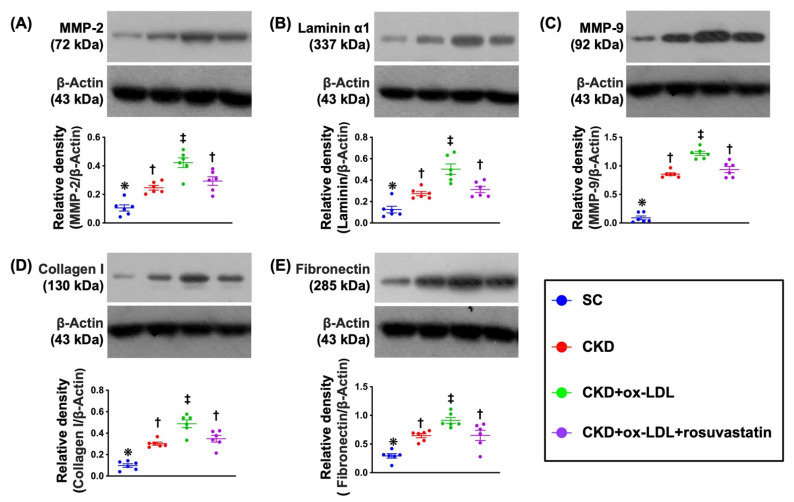
Protein expression of ECM in kidney parenchyma by day 42 after CKD induction. (**A**) Protein expression of matrix metalloproteinase (MMP)-2, * vs. other groups with different symbols (†, ‡), *p* < 0.0001. (**B**) Protein expression of MMP-9, * vs. other groups with different symbols (†, ‡), *p* < 0.0001. (**C**) Protein expression of laminin, * vs. other groups with different symbols (†, ‡), *p* < 0.0001. (**D**) Protein expression of fibronectin, * vs. other groups with different symbols (†, ‡), *p* < 0.0001. (**E**) Protein expression of collagen I, * vs. other groups with different symbols (†, ‡), *p* < 0.0001. All statistical analyses were performed by one-way ANOVA, followed by Bonferroni multiple comparison post hoc test (n = 6 for each group). Symbols (*, †, ‡) indicate significance for each other (at 0.05 level).

**Figure 9 antioxidants-11-02465-f009:**
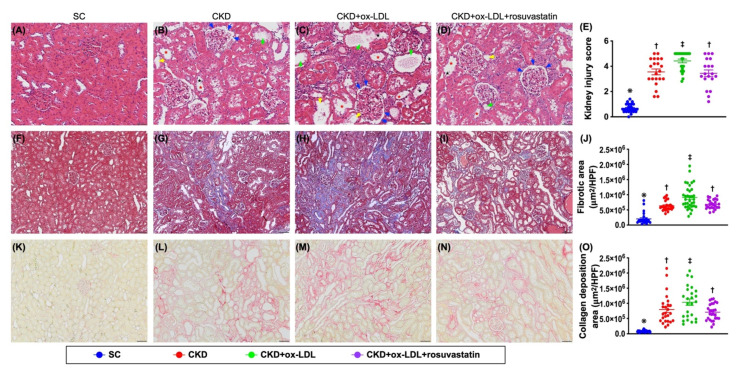
The histopathological analyses of kidney injury score and fibrosis in kidney parenchyma by day 42 after CKD induction. (**A**–**D**) Light microscopic findings (200×; H&E stain) showing significantly increased in loss of brush border in renal tubules (yellow arrows), tubular necrosis (green arrows), tubular dilatation (red asterisk), protein cast formation (black asterisk), and dilatation of Bowman’s capsule (blue arrows) in CKD + oxidized LDL group than in other groups. (**E**) Analytical result of kidney injury score, * vs. other group with different symbols (†, ‡), *p* < 0.0001. (**F**–**I**) Illustrating the microscopic finding (200×) of Masson’s stain for identification of fibrosis (blue color). (**J**) Analytical result of fibrotic area, * vs. other group with different symbols (†, ‡), *p* < 0.0001. (**K**–**N**) Illustrating the histological finding (200×) of Sirius red stain for identification of condensed collagen-deposition area in renal parenchyma (pink color). (**O**) Analytical result of condensed collagen-deposition area, * vs. other group with different symbols (†, ‡), *p* < 0.0001. Scale bars in right lower corner represent 50 µm. All statistical analyses were performed by one-way ANOVA, followed by Bonferroni multiple comparison post hoc test (n = 6 for each group). Symbols (*, †, ‡) indicate significance for each other (at 0.05 level).

**Figure 10 antioxidants-11-02465-f010:**
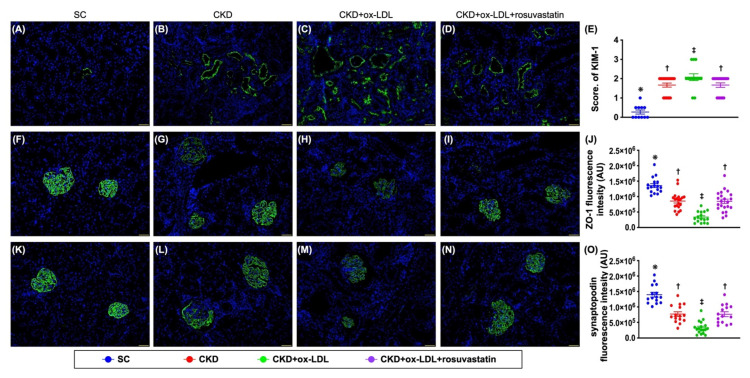
The histopathological analyses of kidney injury molecule and podocyte components in kidney parenchyma by day 42 after CKD induction. (**A**–**D**) Illustrating the immunofluorescent (IF) microscopic finding (200×) for identification of cellular expressions of kidney injury molecule (KIM)-1 (green color). (**E**) Analytical result of the expression of KIM-1, * vs. other group with different symbols (†, ‡), *p* < 0.0001. (**F**–**I**) Illustrating the IF microscopic finding (200×) for identification of cellular expressions of ZO-1 (green color). (**J**) Analytical result of expression of ZO-1, * vs. other group with different symbols (†, ‡), *p* < 0.0001. (**K**–**N**) Illustrating the IF microscopic finding (200×) for identification of cellular expression of synaptopodin (green color). (**O**) Analytical result of expression of synaptopodin, * vs. other group with different symbols (†, ‡), *p* < 0.0001. Scale bars in right lower corner represent 50 µm. All statistical analyses were performed by one-way ANOVA, followed by Bonferroni multiple comparison post hoc test (n = 6 for each group). Symbols (*, †, ‡) indicate significance for each other (at 0.05 level).

## Data Availability

Data are available within the manuscript and Appendix A.

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
