# Peer review of "Oxidized-LDL Deteriorated the Renal Residual Function and Parenchyma in CKD Rat through Upregulating Epithelial Mesenchymal Transition and Extracellular Matrix-Mediated Tubulointerstitial Fibrosis—Pharmacomodulation of Rosuvastatin"

_antioxidants, 2022, doi:10.3390/antiox11122465_

Round 1
Reviewer 1 Report
Development of therapeutic approaches for the preservation of renal function in CKD is urgent and essential. Definite plus of this manuscript is the developed animal model of the aggravated CKD in rats – 5/6 nephrectomy followed by intra-renal arterial administration of oxidized LDL. The described protective features and proposed mechanism(s) of CKD+ox-LDL animals with rosuvastatin are interesting; however, presented results and discussion lack comprehension, therefore, it is not convincing.
Concerns:
1. Abstract is written using very heavy language style with a lot of unnecessary details; at the same time, it lacks comprehension and clear conclusion.
2. Introduction does not present compelling case for studding effects of ox-LDL in CKD.
3. The manuscript is mainly descriptive, lacking mechanistic insights. In general, results (figures) consist of a list of compared features between treatment groups of cells or animals (for ex., proteins expression levels) without any mechanistic approaches. For ex., cell culture experiments showing protective effect of rosuvastatin will be very mechanistically insightful.
Minor concerns:
1. The justification of using only male animals should be included.
2. Labeling of figure panels with WB is confusing. Figs. 1, 4, 8, 10 have double letters. It would be better to present WB results as it done in Fig. 9.
3. Panels in figures with microscopic images and their quantifications should be arranged and labeled more reader friendly. For ex., 5 panels in a row: 4 images with treatment labeling and 1 quantification graph. It applies to Figs. 2-5; 11-12.
4. Figs. 2 and 3 should be combined, they lead to the same conclusion.
5. Fig. 4A-E simply confirms uptake of ox-LDL and should be moved to Supplemental materials as Fig. S1. Fig. 4F-H – is this result from cellular lysates or conditioning media?
6. Fig. 5I – what is “Wound healing (%))? Fig. 5 - the results on cell viability are discussed but not shown.
7. Fig. 6 model is very confusing in presentation.
8. Manuscript should be thorough checked for spelling and grammar. There are some important typos.
9. Discussion is very limited and does not sufficiently and meaningfully discuss data presented in the manuscript. Also, there are numerous references to Fig. 7 in Discussion, which does not make any sense.
Author Response
Response to Specific Comments to Author
Comment 1: Abstract is written using very heavy language style with a lot of unnecessary details; at the same time, it lacks comprehension and clear conclusion.
Response 1: Yes, according to your criticism, we have rewritten the Abstract in our revised manuscript.
Comment 2: Introduction does not present compelling case for studying effects of ox-LDL in CKD
Response 2: Yes, according to your comment, we have provided this information in the last paragraph of Introduction in our revised manuscript.
Comment 3: The manuscript is mainly descriptive, lacking mechanistic insights. In general, results (figures) consist of a list of compared features between treatment groups of cells or animals (for ex., proteins expression levels) without any mechanistic approaches. For ex., cell culture experiments showing protective effect of rosuvastatin will be very mechanistically insightful.
Response 3: Dear reviewer, as you can find that the Figures 6 (revised as new Figure 4) and supplementary Figure 1 clearly illustrated the underlying mechanism of oxidized LDL induced the deterioration of residual renal function in CKD. The proposed mechanism was fundamentally based on the findings of all Figures rather merely one Figure. Actually, you can see that in the Discussion paragraph, we had clearly and step by step discussed our findings which fundamentally constituted the findings in Figure 6 (revised as new Figure 4). Additionally, we had also extensively discussed the role of rosuvastatin on protecting the kidney in CKD rodent in the Discussion paragraph. We recommend that our findings are full of novelty. We wish the reviewer could more focus on the Figure 6 (revised as new Figure 4) and give more appreciation for our hard work and deep investigation in the present study.
Response to Minor Concerns
Comment 1: The justification of using only male animals should be included.
Response 1: The reason why only male animals were utilized in the present study was that the company (i.e., Charles River Technology, BioLASCO Taiwan Co. Ltd., Taiwan) could not provide the female animals for this study. This information has been stated in the Methodology section of our revised manuscript.
Comment 2: Labeling of figure panels with WB is confusing. Figs. 1, 4, 8, 10 have double letters. It would be better to present WB results as it done in Fig. 9.
Response 2: Yes, these figure panels have been rearranged in the Figs. 1, 4, 8 and 10 in our revised manuscript.
Comment 3: Panels in figures with microscopic images and their quantifications should be arranged and labeled more reader friendly. For ex., 5 panels in a row: 4 images with treatment labeling and 1 quantification graph. It applies to Figs. 2-5; 11-12.
Response 3: Yes, according to your recommendation, we have re-edited the expression of images of each Figure in our revised manuscript.
Comment 4: Figs. 2 and 3 should be combined, they lead to the same conclusion.
Response 4: Yes, according to your recommendation, these two Figures have been merged as a new Figure 2 in our revised manuscript.
Comment 5: Fig. 4A-E simply confirms uptake of ox-LDL and should be moved to Supplemental materials as Fig. S1. Fig. 4F-H – is this result from cellular lysates or conditioning media?
Response 5: (1) Yes, this Figure 4 has been removed to supplementary area as Supplementary Figure 3. (2) The results of 4F-H were from cell lysates
Comment 6: Fig. 5I – what is “Wound healing (%))? Fig. 5 - the results on cell viability are discussed but not shown.
Response 6: (1) The description of “Wound healing (%)” has been stated in the Methodology paragraph of our revised manuscript. (2) We sincerely apology for that the description of “Cell viability” on the Methodology section was just a mistake that has been corrected and described as “Assessment of wound healing migratory ability” in our revised manuscript.
Comment 6: Fig. 6 model is very confusing in presentation.
Response 6: Dear reviewer, we are honest to tell you that the Figure 6 (as a new Figure 4 in our revised manuscript) which has been assiduously illustrated by us based on the results of our findings was categorized into three panels, i.e., including upper, middle, and lower panels. The upper panel used the words to fully explain the pathological process of the renal tubular epithelial cells in the middle panel after the oxidized LDL or TGF-β treatment. On the other hand, the lower panel fundamentally concluded the final pathological outcomes of renal tubular epithelial cells after the oxidized LDL or TGF-β treatment. We propose that the readers could easily understand and appreciate the illustration (i.e., like cartoon picture) of the underling mechanism of oxidized LDL or TGF-β induced renal tubular epithelial cells into EMT.
Comment 7: Manuscript should be thorough checked for spelling and grammar. There are some important typos.
Response 7: Yes, according to your recommendation, we have had another native English speaker to edit our revised manuscript again.
Comment 8: Discussion is very limited and does not sufficiently and meaningfully discuss data presented in the manuscript. Also, there are numerous references to Fig. 7 in Discussion, which does not make any sense.
Response 8: Yes, according to your recommendation, we have provided more extensively discussion in the Discussion paragraph of our revised manuscript. Additionally, the inappropriate reference to Figure 7 has been corrected in our revised manuscript.
We are greatly indebted to you for your professional comments.

Reviewer 2 Report
This is a well done study with potentially interesting findings. I have a fewcomments that needsto be addressed:
1. Basic primary readouts of any intervention in CKD are missing and must be added: blood pressure data over time in all groups, data on GFR and urinary albumin excretion for example urinary albumin to creatinine ratio
2. Recent clinical studies do not support a major role for oxLDL for patients with end-stage renal disease (Zuo et al., Head-to-Head Comparison of Oxidative Stress Biomarkers for All-Cause Mortality in Hemodialysis Patients. Antioxidants (Basel). 2022 Oct 2;11(10):1975. doi: 10.3390/antiox11101975.) Discuss these findings in the light of your study.
Author Response
Response to reviewer’s comments (reviewer #2)
Dear Reviewer:
Your constructive criticism is greatly appreciated. We have made the following responses to comply with your honorable suggestions (Note: The revised parts of the manuscript in response to Reviewer’s comments have been marked in pink color):
Response to Comments and Suggestions for Authors
Comment 1: Basic primary readouts of any intervention in CKD are missing and must be added: blood pressure data over time in all groups, data on GFR and urinary albumin excretion for example urinary albumin to creatinine ratio
Response 1: Dear reviewer, as you can find in the Figure 7 (revised as new Figure 5) that we have provided the serial changes of important renal function parameters, including BUN, creatinine and the ratio of urine protein to urine creation. However, we are honest to tell you that we did not measure the time courses of blood pressure in the animals because we thought that rosuvastatin therapy did not alter the blood pressure in the animals. We apology for our mistake. On the other hand, we are also honest to tell you that we did not know how to measure the GFR in the rodent. Accordingly, we are sorry for that we can’t provide this information.
Comment 2: Recent clinical studies do not support a major role for oxLDL for patients with end-stage renal disease (Zuo et al., Head-to-Head Comparison of Oxidative Stress Biomarkers for All-Cause Mortality in Hemodialysis Patients. Antioxidants (Basel). 2022 Oct 2;11(10):1975. doi: 10.3390/antiox11101975.). Discuss these findings in the light of your study.
Response 2: Yes, according to your recommendation, we have discussed the findings from the recent study in the light of our study in the Discussion paragraph of our revised manuscript.
We would like to take this opportunity to express our appreciation for your detailed review of the article and the kindness of giving us valuable suggestions. Thank you very, very much!

Round 2
Reviewer 1 Report
In the revised version of the manuscript, most of this reviewer’s concerns were addressed which improved quality of the manuscript. However, the response to concern #3 is unsatisfactory. The response to minor concern #6 (Fig. 4) should be incorporated into text and Figure legend. In addition, there are more minor concerns from this reviewer that should be addressed.
Additional minor concerns:
1. Fig. S1 – The schematic should be taken out; it is over-estimation of data presented in the figure. At least, E-cad should be eliminated since data simply are not presented.
2. There is no need to put abbreviations’ explanations in each figure legend. It would be enough to have one abbreviations list for the manuscript. Also, no need to put groups description in figure legends; it is already in the figures itself.
3. It needs to be stated what particular type of laminin was assessed in the studies.
4. Manuscript should still be thorough checked for spelling and grammar. There are some important typos.
Author Response
Response to Reviewer #1
Dear Reviewer:
We have made the following responses to comply with your professional suggestions (Note: The revised parts of the manuscript in response to Reviewer’s comments have been marked in red color):
Comments and Suggestions for Authors
Comment 1: In the revised version of the manuscript, most of this reviewer’s concerns were addressed which improved quality of the manuscript. However, the response to concern #3 is unsatisfactory.
Response1: Yes, according to your comment, we have tried our best to comply with your professional comment in our revised manuscript.
Comment 2: The response to minor concern #6 (Fig. 4) has been incorporated into text and Figure legend of our revised manuscript.
Response 2: Yes, according to your recommendation, our response to minor concern #6 (Fig. 4) has been incorporated into text and Figure legend of our revised manuscript.
Comment 3: In addition, there are more minor concerns from this reviewer that should be addressed.
Response 3: Yes, these have been addressed in the following specific response point by point
Response to Additional minor concerns:
Comment 1: Fig. S1 – The schematic should be taken out; it is over-estimation of data presented in the figure. At least, E-cad should be eliminated since data simply are not presented.
Response 1: Yes, according to you recommendation, we have removed the E-cad in Fig. S1 of our revised manuscript.
Comment 2: There is no need to put abbreviations’ explanations in each figure legend. It would be enough to have one abbreviations list for the manuscript. Also, no need to put groups description in figure legends; it is already in the figures itself.
Response 2: Yes, according to your recommendation, we have removed these inappropriate expressions in our revised manuscript.
Comment 3: It needs to be stated what particular type of laminin was assessed in the studies.
Response 3: Yes, the type of the laminin has been stated in Figure 1-G and Figure 8-B of our revised Figures.
Comment 4: Manuscript should still be thorough checked for spelling and grammar. There are some important typos.
Response 4: Yes, we have carefully checked and edited again for our revised manuscript.
Thank you very, very much again for your kindly help!
Reviewer 2 Report
My points were answered adequately. However, I would suggest that all figures with bar graphs be revised. Nowadays it is standard instead of bar graphs to show all measuring points individually, see for example: Chu et al., Biomed Pharmacother. 2022 Sep;153:113357. doi: 10.1016/j.biopha.2022.113357. Figure 2 B -D. Change all bar graphs to this style. This will enable independent recalculation of your data and is thus nowadays the currently used standard in leading journals.
Author Response
Response to Reviewer #2 comment
Dear Reviewer:
We have made the following responses to comply with your professional suggestions.
Comments and Suggestions for Authors
My points were answered adequately. However, I would suggest that all figures with bar graphs be revised. Nowadays it is standard instead of bar graphs to show all measuring points individually, see for example: Chu et al., Biomed Pharmacother. 2022 Sep;153:113357. doi: 10.1016/j.biopha.2022.113357. Figure 2 B -D. Change all bar graphs to this style. This will enable independent recalculation of your data and is thus nowadays the currently used standard in leading journals.
Response: Yes, according to you comment, we have changed all the bar graphs to measuring points in each Figure legend in our revised manuscript.
Thank you again for your professional comment!